# CD8^+^ T Cell Responses during HCV Infection and HCC

**DOI:** 10.3390/jcm10050991

**Published:** 2021-03-02

**Authors:** Maike Hofmann, Catrin Tauber, Nina Hensel, Robert Thimme

**Affiliations:** 1Department of Medicine II, University Hospital Freiburg—Faculty of Medicine, University of Freiburg, Hugstetter Straße 55, 79106 Freiburg, Germany; Catrin.Tauber@uniklinik-freiburg.de (C.T.); Nina.Hensel@uniklinik-freiburg.de (N.H.); 2Faculty of Biology, University of Freiburg, Schänzlestraße 1, 79104 Freiburg, Germany

**Keywords:** HCV, HCC, T cell exhaustion, CD8^+^ T cells, checkpoint blockade therapy

## Abstract

Chronic hepatitis C virus (cHCV) infection is a major global health burden and the leading cause of hepatocellular carcinoma (HCC) in the Western world. The course and outcome of HCV infection is centrally influenced by CD8^+^ T cell responses. Indeed, strong virus-specific CD8^+^ T cell responses are associated with spontaneous viral clearance while failure of these responses, e.g., caused by viral escape and T cell exhaustion, is associated with the development of chronic infection. Recently, heterogeneity within the exhausted HCV-specific CD8^+^ T cells has been observed with implications for immunotherapeutic approaches also for other diseases. In HCC, the presence of tumor-infiltrating and peripheral CD8^+^ T cell responses correlates with a favorable prognosis. Thus, tumor-associated and tumor-specific CD8^+^ T cells are considered suitable targets for immunotherapeutic strategies. Here, we review the current knowledge of CD8+ T cell responses in chronic HCV infection and HCC and their respective failure with the potential consequences for T cell-associated immunotherapeutic approaches.

## 1. Chronic Hepatitis C Virus Infection and Hepatocellular Carcinoma: Major Global Health Burdens Associated with the Liver

Both, chronic hepatitis C virus (cHCV) infection and hepatocellular carcinoma (HCC) affect the liver and represent major global health burdens. Worldwide, more than 71 million people are currently infected with hepatitis C virus (HCV) [1] with estimated 1.75 million new chronically infected patients per year [2]. In more than 70% of cases, acutely infected patients develop a chronic HCV infection. Acute and chronic HCV infection are mostly asymptomatic, however, chronic hepatitis is highly associated with the development of liver fibrosis which can progress to premalignant cirrhotic remodeling of the liver and ultimately to hepatocellular carcinoma [3]. HCC is the most common form of primary liver cancer in adults and is one of the main causes of cancer-related deaths worldwide [4,5,6,7]. By 2040, a further 65% increase in incidence is expected by the WHO [8].

The liver, although no lymphoid organ, has a rich and highly specified immune composition. The liver immune system is normally in a hypoimmune state, guaranteeing balance between tolerance towards harmless molecules and immunity towards pathogens. This state renders the liver susceptible towards infections and cancer [9]. Nevertheless, upon viral infection for example with HCV, the innate immune system is induced with a rapid activation of the interferon response, natural killer cells and a local increase in cytokines and chemokines [10,11]. This is subsequently followed by a delayed infiltration of CD4^+^ and CD8^+^ T cells [12] leading to necro-inflammation. Chronic liver disease associated with chronic necro-inflammation may induce an immunosuppressive, pro-tumorigenic environment [6,13,14] and therefore favors a multifactorial process in which HCC can develop. The tumor microenvironment in HCC consists of various immunosuppressive immune cell populations (e.g., regulatory T cells and myeloid-derived suppressor cells) and immunosuppressive cytokines (e.g., IL-10) [15]. An immunosuppressive tumor microenvironment modulates T cell reactivity [15] and can lead to evasion of HCC from immunosurveillance [16]. Besides chronic viral hepatitis, e.g., induced by cHCV infection, chronic alcohol abuse and non-alcoholic steatohepatosis (NASH), e.g., associated with the metabolic syndrome, frequently drive HCC development also through necro-inflammation. Yet, cHCV infection is still the leading cause of HCC in the Western world [13].

The therapeutic options of HCC are limited and curative therapies such as resection and local ablation are only available for patients with small tumor nodes and well-preserved liver function. Treatment options for patients in advanced stages are mostly restricted to transarterial chemoembolization (TACE), systemic therapy with different agents, or best supportive care due to tumor burden and poor liver function [6]. Thus, risk reduction of HCC development is an important measure in patient care. With respect to chronic viral hepatitis, this is reached by anti-viral treatment. In cHCV infection, the introduction of direct-acting antiviral (DAA) drug therapy in 2014 led to a sustained virological response rate far exceeding 90% of treated patients [17] and thus reduced the risk of HCC development with the exception of patients with undefined/non-malignant hepatic nodules [18,19,20]. Of note, this unique success story of hepatitis C research leading to the development of the highly effective DAA treatment has recently been honored with the Nobel Prize [21]. However, HCV is still far from being eradicated since high costs, limited availability of DAAs, and infrastructural restrictions problems hamper a world-wide campaign [22,23]. Additionally, the incidences of DAA-resistant cases and patient groups with poor prospects for recovery [1] have to be considered. The recent setbacks in vaccine development [24,25,26,27,28] and a lack of immunological protection from reinfection after cure [17,29,30,31] demonstrate that the success of DAA therapy will most probably not be sufficient to reach HCV eradication.

In both, HCV infection and HCC, CD8^+^ T cells constitute major immune effector cells that elicit cytotoxic and non-cytolytic anti-viral and -tumoral effector functions. However, in the context of chronic antigen stimulation, CD8^+^ T cells are often functionally impaired resulting in chronic progression of both liver-associated diseases. Therapeutic approaches that induce CD8^+^ T cells to release their full or at least an improved effector capacity are therefore considered promising in combating the global health burdens by cHCV infection and HCC. This strategy is nicely reflected by the recent attempts of treating HCC by the PD-1 checkpoint inhibitor nivolumab, that among others target CD8^+^ T cells, led to tumor reduction and to a sustained objective response in 15–20% of patients with advanced HCC [32]. A similar response was achieved with the PD-1 checkpoint inhibitor pembrolizumab [33] that also demonstrated a survival benefit after checkpoint blockade therapy in HCC. Moreover, the combination of atezolizumab (anti-PD-1) and bevacizumab (anti-VEGF) in patients with unresectable HCC (including but not stratifying viral and non-viral etiologies) showed a progression-free survival rate of over 15 months that is superior compared to multikinase inhibitor sorafenib [34] and is thus now considered first line therapy for HCC. In this review, we summarize the current knowledge of CD8+ T cell responses in cHCV infection and HCC with a special focus on their respective failure and the open questions since this sets the basis for the design of new or additive therapeutic strategies aiming at improving anti-viral and anti-tumoral CD8^+^ T cell responses.

## 2. CD8^+^ T Cell Responses in HCV

A robust and effective CD8^+^ T cell response in association with a strong support by CD4^+^ T helper cells is necessary for the spontaneous clearance of acute HCV infection [35,36]. Although the occurrence of virus-specific CD8^+^ T cells in the peripheral blood and liver is delayed (6–8 weeks after infection), it is clearly associated with significant reduction of viral load but also with the onset of liver disease [37,38]. Furthermore, CD8^+^ T cell depletion studies conducted in chimpanzees resulted in prolonged viremia, further highlighting the central antiviral role of CD8+ T cells [36]. The central role of CD8^+^ T cell responses is also supported by the observed protective effects of diverse human leukocyte antigen class I alleles, such as HLA-B*27 or –A*03 [39]. Virus-specific CD8^+^ T cells that are associated with HCV clearance in humans are characterized by expression of activation-associated molecules (PD-1, CD39) [38,40], high levels of T-bet [41], and a low cytokine production [37]. After successful viral elimination and subsequent cessation of antigen triggering, the phenotype of HCV-specific CD8^+^ T cells shifts toward a classical memory-associated phenotype (high expression of CD127), inheriting a reduced frequency and the ability to readily expand upon reinfection. This memory of CD8^+^ T cells contributes to an increased likelihood of viral resolution in re-infected patients [42]. In chronic HCV infection, however, the phenotype and functionality of virus-specific CD8^+^ T cells are tremendously altered; the frequencies of virus-specific CD8^+^ T cells are lower and the remaining virus-specific CD8^+^ T cells fail to clear the virus but still most likely contribute to ongoing liver disease [43,44,45].

### Failure of Virus-Specific CD8^+^ T Cells in HCV

Two main mechanisms are associated with the failure of virus-specific CD8^+^ T cells in cHCV infection. One mechanism is the evasion of the virus from the immune system, a phenomenon called viral escape. Viral escape mutations can lead to a loss of HCV-specific CD8^+^ T cell epitopes and loss of de novo T cell priming, contributing to reduced recognition and thus altered anti-viral activity of virus-specific CD8^+^ T cells [40,46,47,48].

The other mechanism is T cell exhaustion, a phenomenon first described in chronic lymphocytic choriomeningitis virus (LCMV) infection in mice [49,50] which can be observed in many chronic viral infections and cancer also in humans. Exhausted virus-specific CD8^+^ T cells have an impaired survival, an altered metabolic, epigenetic, and molecular signature [48,51,52,53,54] and are characterized by a gradual loss of effector functions and proliferative capacity. In particular, exhausted CD8^+^ T cells exhibit an increased expression of co-inhibitory molecules such as PD-1, TIM-3, LAG3, 2B4 [55,56,57], high expression of the transcription factors Eomes [58,59] or the recently identified HMG-box transcription factor TOX [60,61,62,63]. The mechanisms leading to T cell exhaustion are manifold, with high viral load and large number of antigen presenting cells [64,65,66], as well as a loss of CD4^+^ T cell-mediated help and absence of IL-21 signaling playing important roles [50,67,68,69]. Noteworthy, recent studies by the groups of Georg Lauer and Carlo Ferrari showed that transcriptional and metabolic differences in HCV-specific CD8^+^ T cells are already present at early time points of infection and influence the outcome of infection [48,51].

Exhausted virus-specific CD8^+^ T cells are heterogeneous and consist of subpopulations with different functional and phenotypic properties [56,59,60,61,63,70,71,72,73,74]. Based on the expression of the inhibitory receptor PD-1 and the IL-7 receptor α-chain CD127, Wieland et al. could identify CD127^+^ PD-1^+^ memory-like subsets expressing the transcription factor TCF-1 and CD127^-^ PD-1^high^ Eomes^high^ terminally differentiated subsets of exhausted HCV-specific CD8^+^ T cells in cHCV infection. CD127^+^ PD-1^+^ memory-like subsets determine the proliferative capacity of the HCV-specific CD8^+^ T cells population in cHCV infection that can be further appointed to the TCF-1 expression of this subset. Furthermore, only the memory-like subset is maintained independently from antigen recognition either after viral escape or after antigen withdrawal by DAA therapy [75]. These findings have two important implications: first, ongoing antigen recognition drives differentiation towards terminal exhaustion; and second, the memory-like subset maintains the virus-specific CD8^+^ T cell pool during and after cHCV infection. In addition, it has been shown that different exhausted virus-specific CD8^+^ T cell subpopulations in mice and humans are differently responsive towards checkpoint blockade with memory-like CD8^+^ T cells representing the best responders [55,76].

Noteworthy, the memory-like HCV-specific CD8^+^ T cell subset retains characteristics of exhausted T cells even after viral elimination by DAA therapy, like a molecular scar of chronicity, and remains functionally inferior compared to conventional memory HCV-specific CD8^+^ T cells emerging after self-limiting HCV infection [75,77]. Consequently, antigen withdrawal after long-term persisting antigen recognition does not lead to full recovery of exhausted HCV-specific CD8^+^ T cells and thus to an impaired CD8^+^ T cell memory [77]. These findings have important translational consequences as they at least partially explain the limited protective capacity of the HCV-specific CD8^+^ T cells after DAA cure and reinfection [75]. Thus, these results clearly implicate the need to therapeutically target molecular determinants associated with T cell exhaustion to unleash a fully functional and robust CD8^+^ T cell response after DAA-mediated HCV clearance. Further studies are therefore required to fully explore the imprinted molecular and epigenetic signatures in exhausted HCV-specific CD8^+^ T cells including the definition of master regulators associated with the differentiation program of T cell exhaustion. A first step in this direction was the recent identification of the HMG-box transcription factor TOX that regulates the epigenetic and transcriptional program in exhausted T cells in mouse models of chronic LCMV infection and cancer [60,61,62] and that is also associated with the exhausted phenotype of HCV-specific CD8^+^ T cells [60] and PD1^+^ T cells in HCC [78]. Moreover, this observation of an association of TOX with exhausted T cell characteristics in cHCV and HCC highlights shared principles of CD8^+^ T cell dysfunction in both liver-associated diseases.

## 3. CD8^+^ T Cell Responses in HCC

HCC patients with detectable antigen-specific CD8^+^ T cell responses during the natural course or induced by anti-tumoral therapy have an improved overall survival. Indeed, lymphocyte infiltrates, in particular tumor-infiltrating CD8^+^ T cells, have been associated with improved survival and lower relapse rates after liver resection [5,79,80,81,82,83]. However, HCC tumors are frequently only poorly infiltrated by CD8^+^ T cells or other immune cells [5].

HCC is characterized by a high molecular complexity and genetic heterogeneity, leading to the identification of different tumor antigens in HCC tissue (as well as in other cancers) [84,85,86,87,88,89,90,91,92]. Tumor antigens were initially identified in other tumor types and subsequently examined for their expression and immunogenicity in HCC [93]. Tumor antigens are classified in tumor associated antigens (TAA) and tumor-specific mutant antigens (neoantigen). Neoantigens are rare and only a few have been so far identified in the context of HCC [94,95,96]. In contrast, TAA were detectable in the HLA ligandomes of HCC patients [94,95,96]. With respect to TAA, several categories are distinguished based on the expression pattern, namely tumor testis antigens, overexpressed antigens, differentiation antigens, oncofetal antigens. Another category of antigens in the context of HCC are viral antigens [16,92] (Figure 1). Although TAA may be partially subject to self-tolerance, they are considered as good targets for immunotherapeutic treatment concepts, because they are mainly expressed in tumor cells and shared between patient groups. Of note, differential TAA expression in HCC with different underlying etiologies has not been extensively addressed so far.

TAA-specific CD8^+^ T cells, which are able to recognize respective tumor antigens in autologous tumor tissue, have been shown to be associated with tumor control. For example, the study of Flecken et al. showed that CD8^+^ T cell responses specific for TAA are associated with prolonged progression-free survival in HCC patients [84]. The expression of HLA class I molecules in primary HCC [97] and several groups of TAAs that enable CD8^+^ T cells to recognize tumor cells have been identified [15,84,92]. However, the expression rates of TAA are different. For example, MAGE-A is expressed in up to 80% of HCC patients, while NY-ESO-1 is expressed in less than 50% of HCC patients [98,99,100]. Indeed, several studies showed a significant heterogeneity and no consistent hierarchy between different TAA-specific CD8^+^ T cell responses within individual cohorts of HCC patients [84,85,98]. Taken together, the relevant immunodominant epitopes targeted by CD8^+^ T cells are so far not well understood in HCC contrary to well defined viral CD8^+^ T cell epitopes in cHCV infection [86,101].

### Failure of CD8^+^ T Cells in HCC

Although TAA-specific CD8^+^ T cells are associated with a better outcome [84], there is evidence of a dysfunctional state of these cells [99]. The underlying molecular mechanisms are, however, largely unknown due to the low frequency of TAA-specific CD8^+^ T cells in HCC patients. Thus, most studies are based on in vitro expansion protocols with cytokines such as IL-2 or IL-12 [16] prior to ELISPOT assays, limiting the analysis of the ex vivo molecular properties of TAA-specific CD8^+^ T cells in HCC. By applying peptide/MHCI tetramer-based enrichment, we characterized circulating TAA-specific CD8^+^ T cells targeting gylpican-3, NY-ESO-1, MAGE-A1, and MAGE-A3 in therapy-naïve HCC patients [102]. The frequencies of TAA-specific CD8^+^ T cells were comparable in HCC patients and in healthy donors (HD) or in patients with liver cirrhosis but lower compared to virus-specific CD8^+^ T cells present in HCC patients. Significantly more TAA-specific CD8^+^ T cells from HCC patients (expressing the respective TAA) displayed an antigen-experienced phenotype (% antigen-experienced MAGE-A-specific CD8^+^ T cells in HCC: Median: 52.9%; IQR: 60.8%) [102]. This observation indicates an at least partially inefficient TAA-specific CD8^+^ T cell priming and activation, which may lead to limited expansion and thus to low frequencies that are barely distinguishable from the naïve precursor frequencies. Virus-specific CD8^+^ T cells in the very same patients showed an antigen-experienced phenotype, which contrasts with an actively persisting general cancer-associated mechanism of improper T cell priming in HCC.

In general, as discussed above, persistent antigen recognition leads to a gradual exhaustion of CD8^+^ T cells in chronic viral infections and cancer [103,104]. Surprisingly, and in contrast to HCV-specific CD8^+^ T cells in cHCV patients, antigen-experienced MAGE-A-specific CD8^+^ T cells do not show a terminally exhausted phenotype (Eomes^hi^, PD-1^hi^, TCF-1^lo^, CD127^lo^) in therapy-naïve HCC patients [102], potentially reflecting different quantities and qualities of antigen recognition. In addition, only moderate expression of multiple inhibitory receptors—also characteristic marker molecules of T cell exhaustion—has been shown on TAA-specific CD8^+^ T cells targeting several TAAs in different cohorts of HCC patients with different underlying etiologies [98,101,105,106]. A higher expression of these inhibitory receptors was observed on TAA-specific CD8^+^ T cells isolated from HCC tissue compared to T cells from tumor-free liver tissue or blood, especially a significantly higher PD-1 expression was detectable on tumor-infiltrating lymphocytes compared to PBMC [98,101,105,106]. Blocking antibodies targeting these inhibitory receptors (PD-1, TIM3, and LAG3) restored T cell function and combinations of antibodies had additive effects [101]. This observation suggests that the even moderate expression of inhibitory receptors on TAA-specific CD8^+^ T cells dampens the T cell response in HCC. However, further investigations are required to clarify the molecular signatures and with this the exhausted state of TAA-specific CD8^+^ T cells also in relation to the underlying etiology. Interestingly, CD8^+^ T cell subsets with molecular signatures of T cell exhaustion (including PD-1 expression) were identified by single cell RNA sequencing of bulk T cells isolated from peripheral blood and HCCs [7]. Although the targeted antigens (including TAAs, neoantigens, and viral antigens) of these exhausted CD8^+^ T cells remain elusive this finding provides a mechanistic explanation of the durable objective response to PD-1 checkpoint blockade therapy of at least some patients with advanced HCC [32,107]. Further studies are now required to define the tumor antigens targeted by these exhausted CD8^+^ T cells and subsequently to more precisely analyze their specific molecular profiles in order to optimize checkpoint therapies and other immunotherapeutic approaches like vaccination strategies. Of note, CD8^+^ T cell responses are markedly affected by the tumor microenvironment (TME). This notion is best exemplified by (i) a study of Di Blasi et al. showing that the presence of certain TIL clusters (e.g., ICOS^+^ TIGIT^+^ CD4^+^ TILs) can serve as a prognostic indicator for the response to checkpoint blockade therapy [108]; and by (ii) the improved progression-free survival of HCC patients treated with Atezolizumab (blocking PD-1 checkpoint pathway) in combination with the VEGF-blocking antibody Bevacizumab (blocking VEGF) that has recently been approved as first line therapy in HCC [34]. Thus, a deep understanding of the overall immune contexture including tumor-resident and tumor-specific immune cells is crucial to answer the following important questions also in relation to the design of new or improved immunotherapeutic approaches: Which immune cells primarily respond to immunotherapy?; Which of the responding immune cells are beneficial, which are deleterious in the anti-tumor response? Or which immune cells support the anti-tumor CD8^+^ T cell response? Why are only some tumors accessible for immunotherapy?; and which T cell subsets mediate anti-tumoral activity in patients who respond to checkpoint blockade therapy?

## 4. Concluding Remarks

Clearly, CD8^+^ T cells are major effector cells in anti-viral and anti-tumoral immunity in cHCV infection and HCC and CD8^+^ T cell impairment is common in both liver-associated diseases. The approval of DAA therapy not only revolutionized patient care but also provides a unique chance to study CD8^+^ T cell impairment by T cell exhaustion in a clinically relevant setting. By taking this opportunity, recent studies [51,60,75,109,110] underlined the relevant role of ongoing antigen recognition in driving HCV-specific CD8^+^ T cell exhaustion and offered novel insights into phenotypic and functional heterogeneity, metabolic dysregulation and fate of exhausted CD8^+^ T cells during and after cHCV infection. Although there is some data available regarding CD8^+^ T cell exhaustion in HCC [7,101] much less is understood concerning mechanistic details due to the absence of knowledge about the targeted antigens. However, general principles of CD8^+^ T cell exhaustion/dysfunction seem to be conserved between virus-specific and tumor-specific CD8^+^ T cells [103] which is best reflected by the common master regulator TOX [60,61,62,78]. Definition of central factors driving T cell exhaustion/dysfunction that are targetable by immunotherapeutic approaches may therefore be translatable from chronic viral infections to cancer and vice versa. However, there are also considerable differences in chronic viral infections and cancer that potentially impact the CD8^+^ T cell response: e.g., origin of antigen (exogenous versus endogenous) associated with antigen quantity and presentation, the differential expression of cytokines and other immune mediators (pro-inflammatory versus immunosuppressive), and the composition of the other immune cells (CD4^+^ T cell help, CD4^+^ regulatory T cells, myeloid-derived suppressor cells, and tumor-associated macrophages). Thus, future studies are required to define shared and diverging determinants and molecular characteristics of CD8^+^ T cell dysfunction in chronic viral infection and cancer. For this, cHCV infection and HCC represent important translational settings since both affect the liver and may also occur in combination helping to dissect virus- and tumor-associated effects on CD8^+^ T cell dysfunction. This knowledge will provide rationales for establishing predictive biomarkers, e.g., responding/beneficial immune cells and CD8^+^ T cell subsets, and the design of novel or improved immunotherapeutic approaches, like combinatorial treatments lowering inhibitory signals from the microenvironment and recruiting/boosting the most functional CD8^+^ T cell response (Figure 2). Both are especially urgently needed for HCC treatment.

## Figures and Tables

**Figure 1 jcm-10-00991-f001:**
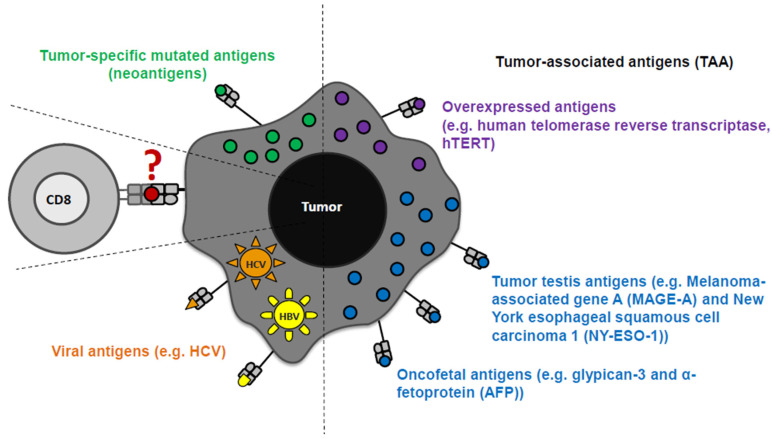
Tumor antigens in the context of hepatocellular carcinoma (HCC). Tumor-specific antigens (neoantigens), tumor-associated antigens (TAA: overexpressed antigens, tumor testis antigens, and oncofetal antigens), and viral antigens are detectable in HCC patients.

**Figure 2 jcm-10-00991-f002:**
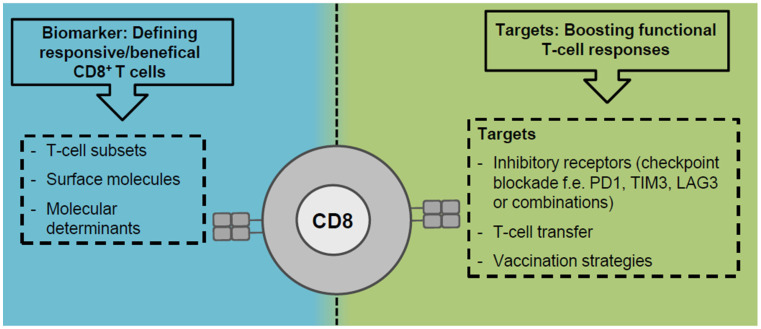
Perspectives of CD8^+^ T cells in immunotherapeutic approaches. CD8^+^ T cells may have potential roles as targets and biomarkers in immunotherapy.

## Data Availability

This review article is based on previously published work.

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
