# Peer review of "CD8+ T Cell Responses during HCV Infection and HCC"

_jcm, 2021, doi:10.3390/jcm10050991_

Round 1
Reviewer 1 Report
- Paper is well conducted and written
- Paragraph 5 is a quite so long and Figure 2 is not so essential
- It is necessary to check many abbreviations (for example: Lines 9-25-28-103-10-26-33-309)
- Review references according to method of this journal for numbers of authors and acronyms
- Optimize legend of Figure 1
Author Response
- Paper is well conducted and written
We thank the reviewer for his/her kind evaluation and his/her valuable comments that helped to improve our manuscript.
- Paragraph 5 is a quite so long and Figure 2 is not so essential
According to the reviewer request, we shortened paragraph 5, deleted the original figure 2 and instead included another figure as also requested by reviewer 2 with the following figure legend: “Perspectives of CD8+ T cells in immunotherapeutic approaches. CD8+ T cells may have potential roles as targets and biomarkers in immunotherapy.”
- It is necessary to check many abbreviations (for example: Lines 9-25-28-103-10-26-33-309)
We checked the use of abbreviations and adapted them in the revised manuscript.
- Review references according to method of this journal for numbers of authors and acronyms
We adapted the references in the revised manuscript.
- Optimize legend of Figure 1
We revised Figure 1 and the figure legend to clarify the classes of tumor antigens in the context of HCC as followed: “Tumor antigens in the context of HCC. Tumor-specific antigens (neoantigens), tumor-associated antigens (TAA: overexpressed antigens, tumor testis antigens and oncofetal antigens) and viral antigens are detectable in HCC patients.
Reviewer 2 Report
The review manuscript entitled “CD8+ T-cell responses during HCV infection and HCC” has been prepared with great care, it reads well and contains useful information. This review is timely and appealing due to the medical needs in HCC cancer biology, and highlights major questions that urgently need to be addressed. Here are some suggestions that can improve the quality of the manuscript.
Major points:
1-The liver plays an important role in the clearance of infectious pathogens, in antigen presentation and in maintaining the balance between immune system activation and immunotolerance. It would be highly beneficial for non-specialized readers to include some information about the immunomodulatory role of the liver and to give a brief overview of the immune cell composition in the normal liver, and in the context of chronic HCV infection and HCC. Moreover, the authors should mention the role of immunoinhibitory molecules, as IL-10 or TGF-ß, contributing to the evasion of HCC from immunosurveillance.
2- The authors highlight some interesting perspectives, but should enriched the part dedicated to the therapeutic options targeting CD8+ T-cells in cHCV and HCC. Also, this should appear in a new figure or be integrated in the existing figure(s).
3- Can the authors further comment on the fact that MAGE-A-specific CD8+ T cells do not show an exhausted phenotype in HCC (lane 246), whereas single cell RNA sequencing identified some CD8+ T cells subsets with T-cell exhaustion molecular markers, consistent with the relative efficiency of checkpoint blockade therapy in HCC.
Minor points
1- Page 3, lane 105, the authors state “In chronic HCV (cHCV) infection, however, the phenotype and functionality of virus-specific CD8+ T cells is tremendously altered; the frequencies are lower and the remaining virus specific CD8+ T cells fail to clear the virus…”. Please specify to what “the frequencies” refer.
2- Page 6, lane 230, the authors write “Significantly more TAA-specific CD8+ T cells from HCC patients displayed an antigen-experienced phenotype compared to HD but some cells still have a naïve phenotype indicating lack of priming or abortive activation”. Please precise the proportion of TAA-specific CD8+ T cells that have a naïve phenotype.
3- Figure 2 is confusing in its shape and message. Moreover, the legend is too short to guide the reader efficiently.
Author Response
The review manuscript entitled CD8+ T-cell responses during HCV infection and HCC” has been prepared with great care, it reads well and contains useful information. This review is timely and appealing due to the medical needs in HCC cancer biology, and highlights major questions that urgently need to be addressed. Here are some suggestions that can improve the quality of the manuscript.
We thank the reviewer for his/her kind words and his/her comments which helped to improve our manuscript.
Major points:
1-The liver plays an important role in the clearance of infectious pathogens, in antigen presentation and in maintaining the balance between immune system activation and immunotolerance. It would be highly beneficial for non-specialized readers to include some information about the immunomodulatory role of the liver and to give a brief overview of the immune cell composition in the normal liver, and in the context of chronic HCV infection and HCC. Moreover, the authors should mention the role of immunoinhibitory molecules, as IL-10 or TGF-ß, contributing to the evasion of HCC from immunosurveillance.
According to the reviewer request we included additional information about the immunomodulatory role of the liver in health, HCV infected and in HCC in line 48-62 “The liver, although no lymphoid organ, has a rich and highly specified immune composition. The liver immune system is normally in a hypoimmune state guaranteeing balance between tolerance towards harmless molecules and immunity towards pathogens. This state renders the liver susceptible towards infections and cancer [1]. Nevertheless, upon viral infection for example with HCV, the innate immune system is induced with a rapid activation of the interferon response, Natural killer cells and a local increase in cytokines and chemokines [2,3]. This is subsequently followed by a delayed infiltration of CD4+ and CD8+ T cells [4] leading to necroinflammation. Chronic liver disease associated with chronic necro-inflammation may induce, an immunosuppressive pro-tumorigenic environment [5,6] [7] and therefore favors a multifactorial process in which HCC can develop. The tumor microenvironment in HCC consists of various immunosuppressive immune cell populations (e.g., regulatory T cells and myeloid-derived suppressor cells) and immunosuppressive cytokines (e.g., IL-10) [8]. An immunosuppressive tumor microenvironment modulates T-cell reactivity [8]. and can lead to evasion of HCC from immunosurveillance [9].”
2- The authors highlight some interesting perspectives, but should enriched the part dedicated to the therapeutic options targeting CD8+ T-cells in cHCV and HCC. Also, this should appear in a new figure or be integrated in the existing figure(s).
We extended the part of therapeutic options targeting CD8+ T cells in several paragraphs as followed:
“Definition of central factors driving T-cell exhaustion/dysfunction that are targetable by immunotherapeutic approaches may therefore be translatable from chronic viral infections to cancer and vice versa.” (line 319-322)
“This knowledge will provide rationales for establishing predictive biomarkers, e.g. responding/beneficial immune cells and CD8+ T cell subsets, and the design of novel or improved immunotherapeutic approaches, like combinatorial treatments lowering inhibitory signals from the microenvironment and recruiting/boosting the most functional CD8+ T cell response (Figure 2). Both are especially urgently needed for HCC treatment.” (line 333-338)
“Thus, these results clearly implicate the need to therapeutically target molecular determinants associated with T-cell exhaustion to unleash a fully functional and robust CD8+ T cell response after DAA-mediated HCV clearance.” (Line 182-185)
“Thus, a deep understanding of the overall immune contexture including tumor-resident and tumor-specific immune cells is crucial to answer the following important questions also in relation to the design of new or improved immunotherapeutic approaches: Which immune cells primarily respond to immunotherapy?; Which of the responding immune cells are beneficial, which are deleterious in the anti-tumor response? Or which immune cells support the anti-tumor CD8+ T-cell response? Why are only some tumors accessible for immunotherapy?; and which T-cell subsets mediate anti-tumoral activity in patients who respond to checkpoint blockade therapy?” (Line 294-302)
3- Can the authors further comment on the fact that MAGE-A-specific CD8+ T cells do not show an exhausted phenotype in HCC (lane 246), whereas single cell RNA sequencing identified some CD8+ T cells subsets with T-cell exhaustion molecular markers, consistent with the relative efficiency of checkpoint blockade therapy in HCC.
Exhausted T cells probably recognize antigens other than MAGE-A or at least MAGE-A specific CD8+ T cells are not terminally exhausted. Single cell analyses have so far only been performed on bulk CD8+ T-cell level. To better clarify, we rephrased the paragraph in the revised manuscript. “Interestingly, CD8+ T cell subsets with molecular signatures of T-cell exhaustion (including PD-1 expression) were identified by single cell RNA sequencing of bulk T cells isolated from peripheral blood and HCCs [10]” (line 277-279).
Minor points
1- Page 3, lane 105, the authors state “In chronic HCV (cHCV) infection, however, the phenotype and functionality of virus-specific CD8+ T cells is tremendously altered; the frequencies are lower and the remaining virus specific CD8+ T cells fail to clear the virus…”. Please specify to what “the frequencies” refer.
We changed the sentence to: “In chronic HCV (cHCV) infection, however, the phenotype and functionality of virus-specific CD8+ T cells is tremendously altered; the frequencies of the virus-specific CD8+ T cells are lower and the remaining virus specific CD8+ T cells fail to clear the virus[…]” (line 129-132).
2- Page 6, lane 230, the authors write “Significantly more TAA-specific CD8+ T cells from HCC patients displayed an antigen-experienced phenotype compared to HD but some cells still have a naïve phenotype indicating lack of priming or abortive activation”. Please precise the proportion of TAA-specific CD8+ T cells that have a naïve phenotype.
In HD 93% of TAA-specific CD8+ T cells displayed a naïve CD45RA+CCR7+ phenotype. In contrast, 47% of TAA-specific CD8+ T cells obtained from HCC patients showed a naïve phenotype, however, with a high inter-individual variability.
We provide this information according to the reported publication in the revised manuscript (line 249-252) “Significantly more TAA-specific CD8+ T cells from HCC patients (expressing the respective TAA) displayed an antigen-experienced phenotype (% antigen-experienced MAGE-A-specific CD8+ T cells in HCC: Median: 52.9%; IQR: 60.8%) [11].
3- Figure 2 is confusing in its shape and message. Moreover, the legend is too short to guide the reader efficiently.
We deleted the original figure 2 and instead included another figure as this was also pointed out by reviewer 1 with the following figure legend: “Perspectives of CD8+ T cells in immunotherapeutic approaches. CD8+ T cells may have potential roles as targets and biomarkers in immunotherapy."
References
- Kubes, P.; Jenne, C. Immune responses in the liver. Annual Review of Immunology 2018, 36, 247-277.
- Dustin, L.B.; Rice, C.M. Flying under the radar: The immunobiology of hepatitis c. Annu Rev Immunol 2007, 25, 71-99.
- Cheent, K.; Khakoo, S.I. Natural killer cells and hepatitis c: Action and reaction. Gut 2011, 60, 268-278.
- Protzer, U.; Maini, M.K.; Knolle, P.A. Living in the liver: Hepatic infections. Nature Reviews Immunology 2012, 12, 201-213.
- Ringelhan, M.; Pfister, D.; O'Connor, T.; Pikarsky, E.; Heikenwalder, M. The immunology of hepatocellular carcinoma. Nat Immunol 2018, 19, 222-232.
- EASL. Easl clinical practice guidelines: Management of hepatocellular carcinoma. Journal of Hepatology 2018, 69, 182-236.
- Shigeta, K.; Hato, T.; Chen, Y.; Duda, D.G. Anti-vegfr therapy as a partner for immune-based therapy approaches in hcc. In Immunotherapy of hepatocellular carcinoma, Greten, T.F., Ed. 2017.
- Breous, E.; Thimme, R. Potential of immunotherapy for hepatocellular carcinoma. J Hepatol 2011, 54, 830-834.
- Mizukoshi, E.; Kaneko, S. Antigen-specific t cell responses in hepatocellular carcinoma. In Immunotherapy of hepatocellular carcinoma, Greten, T.F., Ed. Springer International Publishing AG: 2017; pp 39-50.
- Zheng, C.; Zheng, L.; Yoo, J.K.; Guo, H.; Zhang, Y.; Guo, X.; Kang, B.; Hu, R.; Huang, J.Y.; Zhang, Q., et al. Landscape of infiltrating t cells in liver cancer revealed by single-cell sequencing. Cell 2017, 169, 1342-1356.
- Tauber, C.; Schultheiss, M.; Luca, R.D.; Buettner, N.; Llewellyn-Lacey, S.; Emmerich, F.; Zehe, S.; Price, D.A.; Neumann-Haefelin, C.; Schmitt-Graeff, A., et al. Inefficient induction of circulating taa-specific cd8+ t-cell responses in hepatocellular carcinoma. Oncotarget 2019, 10, 5194-5206.
Reviewer 3 Report
The review by Hoffmann et al. provides the overview of the current understanding of CD8 T cell responses in chronic HCV infection and HCC, highlighting the mechanisms for the failure of these responses. This review covers an important topic in the field with thorough and insightful discussions.
I have following fine points that requires the authors’ attention:
- For Line 71-80, it would be helpful to specify whether those are cHCV-induced HCC or not.
- It would be helpful if authors include discussions on their recent publication at Nature Immunology.
- Line 183-187 and Figure 1 require clarification. In the text, overexpressed antigens and viral antigens are part of TAA, while in the figure, these three seem to be in separate categories.
- It would be helpful to have a couple sentences commenting on TSA in HCC. Are there any known TSA in HCC?
- Are TAA different in cHCV-induced HCC and other types of HCC? Could authors comment on this?
- Are there studies comparing the frequency and phenotype of TAA-specific CD8 T cell responses (e.g. MAGE-A specific CD8 T cells) between cHCV-induced HCC and other types of HCC? Do cHCV-induced HCC and other types of HCC respond similarly to checkpoint blockade treatment?
- Following lines are missing the references: Line 103-107, Line 215-221, Line 226-228, Line 229-231, Line 234-236, Line 238-240, Line 245-247. If they are from unpublished results, it should be stated.
Author Response
The review by Hoffmann et al. provides the overview of the current understanding of CD8+ T cell responses in chronic HCV infection and HCC, highlighting the mechanisms for the failure of these responses. This review covers an important topic in the field with thorough and insightful discussions.
We thank the reviewer for his/her kind words and his/her comments which helped to improve our manuscript.
I have following fine points that requires the authors’ attention:
- For Line 71-80, it would be helpful to specify whether those are cHCV-induced HCC or not.
The study of Finn et al. include HCC-patients with viral (HBV or HCV), non-viral and alcohol-related tumors. Patients with HCV history (negative for HCV RNA) were considered non-infected with HCV. However, in this study, response rates were not analyzed according to the underlying etiology [1]. According to the information of the reported publication, we included the following part in the revised manuscript: “Moreover, the combination of atezolizumab (anti-PD-1) and bevacizumab (anti-VEGF) in patients with unresectable HCC due to viral and non-viral etiologies showed a progression-free survival rate of over 15 months that is superior compared to multikinase inhibitor sorafenib[1] and is thus now considered first line therapy for HCC.” (line 100-104)
- It would be helpful if authors include discussions on their recent publication at Nature Immunology.
We included our recent publication in the discussions in line 174-177 as followed: “Noteworthy, the memory-like HCV-specific CD8+ T cell subset retains characteristics of exhausted T cells even after viral elimination by DAA therapy, like a molecular scar of chronicity, and remains functionally inferior compared to conventional memory HCV- specific CD8+ T cells […].”
- Line 183-187 and Figure 1 require clarification. In the text, overexpressed antigens and viral antigens are part of TAA, while in the figure, these three seem to be in separate categories.
We agree with the reviewer that the figure legend was misleading. Therefore, we have revised figure 1 and the figure legend. Further, we clarified this in the revised manuscript as followed: “With respect to TAA; several categories are distinguished based on the expression pattern, namely tumor testis antigens, overexpressed antigens, differentiation antigens, oncofetal antigens. Another category of antigens in the context of HCC are viral antigens (Figure 1).” (Line 212-215)
- It would be helpful to have a couple sentences commenting on TSA in HCC. Are there any known TSA in HCC?
Tumor-specific mutated antigens (neoantigens) are rare and have not been well identified in HCC patients. In contrast, normal proteins and cancer-testis antigens were detectable in the HLA ligandomes of HCC patients [2-4].
According to the reviewer request, we included this information in the revised manuscript (line 210-212) “Neoantigens are rare and only a few have been so far identified in the context of HCC [2-4]. In contrast TAA were detectable in the HLA ligandomes of HCC patients [2-4].
- Are TAA different in cHCV-induced HCC and other types of HCC? Could authors comment on this?
An interesting question, however, to our knowledge, TAA have not been studied in relation to etiology to date. We included this information now in the revised manuscript as following: “Of note, differential TAA expression in HCC with different underlying etiologies has not been extensively addressed so far”. (Line 218-220)
- Are there studies comparing the frequency and phenotype of TAA-specific CD8+ T cell responses (e.g. MAGE-A specific CD8 T cells) between cHCV-induced HCC and other types of HCC? Do cHCV-induced HCC and other types of HCC respond similarly to checkpoint blockade treatment?
To our knowledge, there are no systematical comparisons of TAA-specific responses comparing viral and non-viral etiologies, so we included this point into the revised manuscript (line 275-277) “However, further investigations are required to clarify the molecular signatures and with this the exhausted state of TAA-specific CD8+ T cells also in relation to the underlying etiology”
Addressing the second part of the question, in the study of Finn et al. responses to checkpoint therapy were investigated in the context of HCC underlying viral and non-viral etiologies however the study did not stratify between the etiologies. We now have included this information in the first paragraph of the revised manuscript (line 100-104) “Moreover, the combination of atezolizumab (anti-PD-1) and bevacizumab (anti-VEGF) in patients with unresectable HCC (including but not stratifying viral and non-viral etiologies) showed a progression-free survival rate of over 15 months that is superior compared to multikinase inhibitor sorafenib [1] and is thus now considered first line therapy for HCC.”
- Following lines are missing the references: Line 103-107, Line 215-221, Line 226-228, Line 229-231, Line 234-236, Line 238-240, Line 245-247. If they are from unpublished results, it should be stated.
We included the references in the revised manuscript.
References
- Finn, R.S.; Qin, S.; Ikeda, M.; Galle, P.R.; Ducreux, M.; Kim, T.-Y.; Kudo, M.; Breder, V.; Merle, P.; Kaseb, A.O., et al. Atezolizumab plus bevacizumab in unresectable hepatocellular carcinoma. New England Journal of Medicine 2020, 382, 1894-1905.
- Lu, L.; Jiang, J.; Zhan, M.; Zhang, H.; Wang, Q.T.; Sun, S.N.; Guo, X.K.; Yin, H.; Wei, Y.; Li, S.Y., et al. Targeting tumor-associated antigens in hepatocellular carcinoma for immunotherapy: Past pitfalls and future strategies. Hepatology 2020, n/a.
- Löffler, M.W.; Mohr, C.; Bichmann, L.; Freudenmann, L.K.; Walzer, M.; Schroeder, C.M.; Trautwein, N.; Hilke, F.J.; Zinser, R.S.; Mühlenbruch, L., et al. Multi-omics discovery of exome-derived neoantigens in hepatocellular carcinoma. Genome Medicine 2019, 11, 28.
- Dong, L.Q.; Peng, L.H.; Ma, L.J.; Liu, D.B.; Zhang, S.; Luo, S.Z.; Rao, J.H.; Zhu, H.W.; Yang, S.X.; Xi, S.J., et al. Heterogeneous immunogenomic features and distinct escape mechanisms in multifocal hepatocellular carcinoma. J Hepatol 2020, 72, 896-908.
Round 2
Reviewer 2 Report
The authors have convincingly addressed all my concerns.